# Interventional and Surgical Treatments for Pulmonary Arterial Hypertension

**DOI:** 10.3390/jcm10153326

**Published:** 2021-07-28

**Authors:** Tomasz Stącel, Magdalena Latos, Maciej Urlik, Mirosław Nęcki, Remigiusz Antończyk, Tomasz Hrapkowicz, Marcin Kurzyna, Marek Ochman

**Affiliations:** 1Silesian Centre for Heart Diseases in Zabrze, Department of Cardiac, Vascular and Endovascular Surgery and Transplantology, Medical University of Silesia, 40-055 Katowice, Poland; latos.magdalena93@gmail.com (M.L.); maciek.urlik@gmail.com (M.U.); mnecki@sccs.pl (M.N.); welik@wp.pl (R.A.); hrapcio@poczta.onet.pl (T.H.); 2European Health Centre Otwock, Centre of Postgraduate Medical Education, Department of Pulmonary Circulation, Thromboembolic Diseases and Cardiology, 05-400 Otwock, Poland; marcin.kurzyna@ecz-otwock.pl

**Keywords:** pulmonary arterial hypertension, interdisciplinary approach, review, idiopathic pulmonary arterial hypertension

## Abstract

Despite significant advancements in pharmacological treatment, interventional and surgical options are still viable treatments for patients with pulmonary arterial hypertension (PAH), particularly idiopathic PAH. Herein, we review the interventional and surgical treatments for PAH. Atrial septostomy and the Potts shunt can be useful bridging tools for lung transplantation (Ltx), which remains the final surgical treatment among patients who are refractory to any other kind of therapy. Veno-arterial extracorporeal membrane oxygenation (V-A ECMO) remains the ultimate bridging therapy for patients with severe PAH. More importantly, VA-ECMO plays a crucial role during Ltx and provides necessary left ventricular conditioning during the initial postoperative period. Pulmonary denervation may potentially be a new way to ensure better transplant-free survival among patients with the aforementioned disease. However, high-quality randomized controlled trials are needed. As established, obtaining the Eisenmenger physiology among patients with severe pulmonary hypertension by creating artificial defects is associated with improved survival. However, right-to-left shunting may be harmful after Ltx. Closure of the artificially created defects may carry some risk associated with cardiac surgery, especially among patients with Potts shunts. In conclusion, PAH requires an interdisciplinary approach using pharmacological, interventional, and surgical modalities.

## 1. Introduction

Pulmonary arterial hypertension (PAH) is associated with poor prognosis despite recent advancements in pharmacological treatments, as there is no curative therapy [1,2]. Oxygen therapy should be considered in patients who develop hypoxemia [3,4]. Oxygen saturation should be maintained above 90% at rest [4]. Advanced treatment for PAH is directed at the pulmonary arterioles and includes prostanoids, endothelin receptor antagonists, phosphodiesterase 5 inhibitors, soluble guanylate cyclase stimulators, and in some cases calcium channel blockers. It is believed that no more than 5% of PAH patients may benefit from calcium channel blockers during long term treatment [5]. Patients with idiopathic, heritable, and drug-related PAH should undergo vasoreactivity testing during right heart catheterization because such tests facilitate drug selection [3,6]. Despite such treatment, severe pulmonary hypertension can still lead to hemodynamic impairment due to right ventricular (RV) dysfunction. Interventions such as atrial septostomy (AS), use of the Potts shunt, and pulmonary artery denervation (PADN) could be applied as bridging therapy for thoracic transplantations, being the ultimate surgical treatment for such diseases. Moreover, they can be used as palliative care tools [7]. 

Despite significant advancements in pharmacological treatment, interventional and surgical options are still a viable treatment for patients with IPAH and other diseases from group 1 of pulmonary hypertension. Herein, we review the interventional and surgical treatments for PAH.

## 2. Right-to-Left Shunting

Pulmonary hypertension increases vascular resistance to the point of RV systolic dysfunction. The left ventricle in patients with severe forms of PAH can present with a proper ejection fraction. However, pathologically reduced pulmonary flow significantly decreases preload, which ultimately deconditions this ventricle. The overall pathophysiology of cardiac dysfunction among patients with PAH is associated with impaired systemic blood flow, oxygen transport, and passive venous blood stasis [8]. Due to these changes, advanced pulmonary vessel diseases lead to multiorgan failure and death. At this point, obtaining the Eisenmenger physiology among patients with severe pulmonary hypertension is the goal, as it is associated with longer life expectancy and more stable hemodynamics [9]. Procedures involving right-to-left shunting provide decompression of the right ventricle, and they may increase the systemic blood flow through increasing the left ventricle’s preload. However, such treatment will lead to a decrease in oxygen saturation; hence, this method is not suitable for all patients [10]. Providing such treatment is well studied among children with idiopathic PAH (IPAH), as a significant scarcity of donors with matching anthropometrics is observed among this group of potential thoracic organ recipients [11]. As the Potts shunt and AS aim to transfer suprasystemic pulmonary hypertension to Eisenmenger physiology, it is debatable which procedure provides such an effect in a better way. A modeling study involving the CircAdapt cardiovascular model of a patient with PAH of various severities was conducted by Delhaas et al. [12]. They reported the theoretical superiority of Potts shunting over AS. However, these theoretical studies were not supported by randomized controlled trials, as we did not find any in our literature review. Detailed depictions of AS and Potts shunting are presented in Figure 1.

### 2.1. Atrial Septostomy

The rationale for AS comes from the observation that patients with concomitant PAH and patent foramen ovale present a survival advantage over those with PAH, but they lacked the aforementioned congenital heart defect variant. AS allows for right-to-left shunting when pressure in the right atrium is elevated [10,13]. There are several transcatheter and surgical methods for performing AS [14]. Two main techniques are used to perform AS. Stepwise balloon dilatation is considered the procedure of choice. This procedure is shown in Figure 1(1). Puncture of the interatrial septum is performed using a Brockenbrough needle followed by progressively increased balloon catheters until a 10% decrease in the arterial oxygen saturation is noted or an increase of up to 18 mmHg of the left ventricular (LV) end-diastolic pressure is present. Another procedure is blade balloon AS, which is performed with a 5–15 mm blade that cuts the limbs of the oval fossa as it is directed away from the anterior aortic root [14].

Studies have shown that performing AS among patients refractory to pharmacologic treatment is associated with an improved cardiac index (range of advancement: 15–58% from 2.04 ± 0.69 L/min/m^2^ to 2.62 ± 0.84 L/min/m^2^
*p* = 0.0001), New York Heart Association (NYHA) functional class (from 3.49 ± 0.6 to 2.1 ± 0.7), and exercise tolerance [10,13]. In the early follow-up period, 88% of survivors reported significant improvement in syncope, exercise tolerance, and right heart failure (RHF). Sixteen percent of patients undergoing AS became recipients of lung transplantation (Ltx). The mean duration of survival after AS (excluding procedural deaths) was 63.1 months. As reported by Keogh et al., even though such a procedure does not necessarily decrease the mean pulmonary artery pressure (mPAP), it can still improve the functional class and certain hemodynamic parameters [10]. Thorough analysis was presented at the Dana Point meeting, where 223 cases of patients undergoing AS were included (81% of them had IPAH). The main finding supports the safety of AS combined with pharmacological therapy [15]. Such treatment also has its drawbacks, as the procedure itself can be lethal. It was reported that such a procedure has 13–16% immediate mortality, with refractory hypoxemia being the most common cause of death. Careful consideration is advised when it comes to patient selection, especially for patients with end-stage heart failure. Patients with oxygen saturation less than 90%, severe RV failure on cardiorespiratory support, low cardiac output, and high right atrial pressure (mean >20 mmHg) did not benefit from such treatment [10,13]. These are believed to be definite contraindications of this procedure. A long-lasting effect was also not noted among all survivors, as some of them (4 of 15–26.6%) developed closure of the created defect [13,16,17]. The patency of AS can be improved by inserting stents or special devices, such as atrial flow regulators (AFRs), presented in Figure 1(2a,2b).

### 2.2. Atrial Flow Regulator

The concept of using an AFR (Figure 1(2a,2b)) among patients with PAH comes from papers pertaining to left heart failure, as depicted in the AFR-PRELIEVE trial assessing the effect of left-to-right shunting [18]. AFRs in the atrial septum were utilized among patients with heart failure, where the implantation success rate and three-month device patency were both 100% among 36 patients. It was assessed that the right-to-left flow through AS protected with an AFR would be useful among patients refractory to pharmacological treatment due to PAH. Rajeshkumar et al. reported that AFR implantation among patients with severe PAH was a safe and feasible treatment, with an implantation success rate of 100% without any major complications [19]. AFR implantation provided improvement of the cardiac index (from 2.36 ± 0.52 to 2.89 ± 0.56 L/min/m^2^, *p* < 0.01) and systemic oxygen transport (from 367.5 ± 75.5 to 428.0 ± 67.1 mL/min/m^2^, *p* = 0.04) despite a reduction in the oxygen saturation. Among 100% of participants, it relieved syncope and significantly improved the 6-min walk distance (6MWD) (from 377.3 ± 33.2 to 423 ± 31.32 m, *p* = 0.008). Inferior cava vein congestion and pericardial effusion were also reduced [19]. AS followed by AFR implantation can be a feasible way to bridge Ltx among patients with severe pulmonary hypertension on the maximal available pharmacological treatment. This was the treatment of the first Polish patient who underwent AFR implantation [20] and became a double lung recipient in our center shortly thereafter [21].

Butterfly stents have also been described as a useful interatrial septal flow regulator that maintains the patency of AS and alleviates the symptoms of RHF [22]. Thanks to this device, a reduction in the serum brain natriuretic peptide (BNP) concentration was observed in the long-term follow-up (from 490 to 82 ng/mL at 18 months after the procedure), and there was noticeable improvement in the 6MWD (from 80 to 426 m) [22]. However, it is important for such procedures to be performed in experienced centers. Atrial septal stents should be placed through a separate puncture among patients who have previously undergone either balloon dilation or blade septostomy. It is important that the major risk of embolization is associated with the deployment of stents into dilated orifices. Thrombus formation on properly placed stents rarely occurs when low-dose aspirin is administered as a preventative measure [14]. Butterfly stents as a treatment for severe PAH can also serve as a bridge to Ltx, as Prieto et al. described the case of a patient who showed improved cardiac output and renal function after implantation of this device up to the point of becoming eligible for Ltx [23]. The butterfly stent was removed during a complex surgery involving bilateral Ltx.

One of the largest single-center stent-secured AS series (68 patients) was reported by Gorbachevsky et al. [24]. The authors stated that stent-secured AS is more effective if it is used in the group of “intermediate-risk” patients than in the group of “high-risk” patients according to the criteria of the 2015 European Society of Cardiology/European Respiratory Society guideline: lower mortality rate, longer 6MWD, lower BNP level, and longer Ltx-free period or Potts shunt. Additionally, the authors reported that the results in the high-risk group were very good and significantly improved. The 13-year follow-up showed that the AFRs remained in good location and, very importantly, were still patent. The obtained results support the hypothesis that the earlier the intervention is performed, the better the results obtained in terms of prognostic factors. This method can be called a bridge to transplantation, the use of a new pharmacological method, or obtaining an optimal response to its application [25].

### 2.3. The Potts Shunt and Its Modifications

As AS has the ability to improve cardiac dysfunction, it is associated with poor oxygen delivery to crucial organs, e.g., the brain and heart. The Potts shunt is a surgical intervention that creates anastomosis between the left pulmonary artery and descending aorta, which naturally leads to decreased oxygen saturation of the lower extremities [11]. This method is shown in Figure 1(3a). Its implementation among children with advanced PAH significantly enhanced the improved systolic function of the right ventricle, right ventricle–pulmonary artery coupling, and functional status, and prolonged mid-term transplant-free survival [26]. Median tricuspid annular plane systolic excursion (TAPSE) increased by 9.1% (from 11.5 (10.4–12.4) to 12.6 (11.7–13.8), *p* = 0.03). RV fractional area change [FAC%] increased by 84% (from 12.1% (9.1–14.5) to 22.2% (16.9–24.2), *p* = 0.008). It was noted that 72% of the patients were weaned off the intravenous pulmonary vasodilator after the procedure. Such connections supply the brain and heart with blood with better oxygen saturation than AS. The Potts shunt was applied as a treatment for children with severe IPAH, as described by Baruteau et al. [11]. All children in the study group survived the procedure. However, two of them died during the short-term follow-up due to low cardiac output, which was caused by premature withdrawal of PAH-specific therapy. All survivors showed improved cardiac function (a decrease in BNP levels from 608 ± 109 to 76 ± 45 pg/mL; *p* < 0.035) with diminished syncope and other RHF symptoms. Enhancement of the 6MWD was also reported after the procedure (302 ± 95 versus 456 ± 91 min; *p* = 0.038), as well as a reduction in the NYHA class (from IV to I *p* = 0.002). What is also important for such patients, as some of them will require solid organ transplantation, is that all of the children caught up to the normal growth curves [11]. In addition, a significant reduction in IPAH-specific pharmacological treatment was observed. In order to work properly, the Potts shunt requires suprasystemic values of the pulmonary pressure and valves of the right heart to function properly, as shunting occurs between the aorta and pulmonary artery. This is a notable difference in comparison with the AS procedure, when atrial shunting is possible as permitted by tricuspid valve regurgitation [10]. There are certain ways to perform Potts shunting. Such anastomosis can be produced surgically and by the transcatheter approach. There is also a risk of flow reversal with the use of traditional Potts shunts. Therefore, novel therapies involving a unidirectional valve conduit have been developed (Figure 1(3b)). Early experience with such a shunt was described by Rosenzweig et al. [27]. Their study assessed five patients with end-stage IPAH before and after implantation of the aforementioned device. In all patients, there was a noted improvement in their cardiac function (mean NT-proB-type BNP level at baseline versus [vs.] postoperatively: 4013 vs. 1107 ng/mL; RV systolic pressure at baseline vs. postoperatively: from 121 ± 12 to 102 ± 8 mmHg, *p* = 0.04; FAC baseline vs. postoperatively: from 22% to 35%) and they remained transplant-free at the time of the follow-up (range, 3–33 months; median, 6 ± 11 months) [27]. Longer-term follow-up of a larger study group is needed. Although patients were able to reduce their PAH-specific treatment, it is worth mentioning that the implanted device requires them to use aspirin and warfarin, which is also associated with adverse effects. Another modification was proposed by Sayadpour, who suggested that radiofrequency perforation during the creation of anastomosis itself may increase the safety of the transcatheter Potts shunt. His paper stated that no death occurred in their animal trials with this technique. He also explained that as radiofrequency has the potential to coagulate the tissue, it may prevent extravasation of blood at the puncture sites and the created pathway [28].

## 3. Pulmonary Artery Denervation

Studies pertaining to the pathogenesis of IPAH have addressed the imbalance of the vasoconstrictors and vasodilators present in the pulmonary circulation. This approach is the basis of approved pharmacological treatment because the effects of prostacyclin, endothelin, and cGMP-specific phosphodiesterase type 5 on the pulmonary vessels are well known. However, other studies have investigated the involvement of neural reflexes in the development of PAH. Baroreceptors and sympathetic nerve fibers are localized in or near the bifurcation area of the main pulmonary artery [29]. In addition, the evidence that PAH is associated with overstimulation of the sympathetic nervous system and excessive activation of the renin-angiotensin-aldosterone system seems convincing [1,30]. The idea of ablation of the neural fibers and/or receptors within the arterial walls was first introduced as a means of treatment in patients with refractory systemic arterial hypertension [31]. Renal artery denervation causes a significant reduction in systemic blood pressure without post-procedural impairment of kidney function. The first animal trial concerning PADN was performed on dogs by Chen et al. [32]. The investigators concluded that endovascular radiofrequency ablation of baroreceptors and/or sympathetic fibers located near the bifurcation of the main pulmonary artery abolished the pressure response to pulmonary artery pressure increase caused by balloon occlusion of the distal vessels. Their study also reported that PADN downregulates the excessively activated renin–angiotensin–aldosterone system. Although this study had its limitations, it indicated the feasibility and safety of the procedure [32]. Huang et al.’s study described transthoracic PADN (TPADN) in male Sprague Dawley rats [33]. They also assessed the abundance of sympathetic nerves (SN) in the adipose and connective tissue surrounding the main pulmonary artery of rats with a distribution larger than their healthy counterparts. TPADN decreased the mean pulmonary artery pressure in the studied population and improved the hemodynamic parameters [33].

Human trials began in a PADN-1 study that reported that denervation of the pulmonary artery performed intravascularly at the bifurcation of the main pulmonary artery and its nearest division among patients with IPAH provided significantly improved exercise tolerance and significantly decreased the mean pulmonary artery pressure in comparison to the control group [29]. All patients with PADN were able to discontinue the PAH-specific medications, and their functional status after the procedure made them too functionally well to be considered for Ltx. This was phase I of the study. Their research continued as phase II, involving 66 patients with pulmonary hypertension of various causes [34]. It revealed that PADN was associated with significantly improved hemodynamics, exercise tolerance (mean improvement of 94 m in 6MWD p), and less frequent PAH-related events including death at 1 year after the procedure. PAH-related events occurred within the first year among 15% of participants (10 patients). 60% of them reported worsening of symptoms [34]. PADN itself was not associated with an increased risk of death. Further studies with proper randomization and control groups are required as concerns were raised about the conduct of PADN-1 [35]. As a novelty, the PADN procedure has gained increasing attention recently. It has also been studied as a therapeutic method for patients with combined pre-capillary and post-capillary pulmonary hypertension [36]. This treatment also improved the hemodynamic parameters and exercise tolerance. However, further clinical trials are required. The progress of the studies pertaining to this treatment method was carefully assessed by Kim et al. [37]. Their review concluded that even though trials published so far have limitations and a short-term follow-up, their results show the usefulness of such a procedure in patients with pulmonary hypertension refractory to available treatment methods. As evidence for excessive beta1-adrenergic SN and renin-angiotensin-aldosterone system activation among patients with various forms of PH is convincing, there have been studies addressing the use of potential pharmacological treatments [30].

## 4. Surgical Modalities

### 4.1. Right Ventricle Assist Device

Use of an RV assist device (RVAD) among patients with severe pulmonary hypertension would have its rationale as the high mortality rate associated with this disease is associated with RHF. Punnoose et al. described an analysis of the theoretical cardiovascular model of a patient with severe PAH. Their simulation was based on RVAD implantation (Synergy TM continuous flow micro-pump CircuLite Inc., Saddle Brook, NJ) [38]. This study revealed that this device may be helpful for such patients. There are two methods of RVAD implantation, as depicted in Figure 2 (either RV sourcing 2-1 or RA sourcing 2-2). Both shifted RV pressure-volume loops leftward, proving that RV unloading may be possible with the use of either device. As pulmonary hypertension leads to RV failure, mechanical support reduces stroke volume. A decrease in RV loading should improve septal motion and LV diastolic filling. RVAD support also had an effect on LV pressure–volume loops, as those were shifted rightward towards higher end-diastolic volumes, which resulted in an improved stroke volume and aortic pressure. Such an effect was noticeable regardless of sourcing. Various flow rates and additional simulations such as the presence of AS and tricuspid valve regurgitation were also assessed [38]. However, the feasibility of such treatments must be supported by clinical trials or at least case reports. Rosenzweig et al. reported the case of a patient with PAH who was disqualified from thoracic organ transplantation due to advanced secondary organ failure [39]. RVAD was introduced in a 42-year-old patient to improve LV filling and cardiac output. The patient was treated with percutaneous RVAD followed by a durable RVAD for six days until she died of septic shock after pneumonia developed. During the follow-up, her RVAD was functioning properly, and her hemodynamic parameters improved. Another case report pertaining to the use of Impella RP (Abiomed, Danvers, MA, USA) as an assist for a 37-year-old patient with pulmonary hypertension concomitant with mixed connective tissue disease was presented by Hansen et al. [40]. The Impella RP was deployed as a full assist because of profound cardiogenic shock due to acute RHF and sepsis, providing a flow rate of 4.5 L/min. The scheme of this device is shown in Figure 2(3). The echocardiographic examination revealed a modest improvement in the RV diastolic diameter and area (from 5.4 to 5.0 cm and from 31 to 27 cm^2^, respectively). The day after implementing such a device, normalization of extremely increased lactate levels was noted, and the level of aminotransferase progressively decreased over 10 days. The patient’s cardiac condition improved to the point of weaning off the device. Unfortunately, the patient died of gastroesophageal complications including sepsis and an underlying disease. This case also proves that the Impella RP could be an alternative to standard RVAD, especially among patients in dire need of such support.

RV support devices can be considered a destination therapy as well as a bridge to thoracic organ transplantation among patients with PAH, as described by Oh et al. [41]. Their case described a 64-year-old patient who was qualified from treatment with Ltx owing to idiopathic pulmonary fibrosis with PH (group 3: PH due to lung disease). He was already being bridged by veno-venous (VV) extracorporeal membrane oxygenation (ECMO) when he developed severe RHF. As veno-arterial ECMO in this case was assessed as unfavorable, he was bridged to successful Ltx by an RVAD combined with an oxygenator. It is worth mentioning that this case described a patient with pulmonary hypertension related to lung fibrosis (group 3); hence, the outcome of such treatment among patients with IPAH remains unclear.

### 4.2. Aria CV PH System

Many advanced devices have been investigated in recent years. One of the latest developments is called the Aria CV PH System. The safety and effectiveness of this device is to be determined in the study (prospective, non-randomized, single- arm, multi-center; ClinicalTrials.gov Identifier: NCT04555161) that started in March 2021 under the acronym ASPIRE PH [42]. The study completion date is estimated to be in June 2023. The main element of the device is a gas filled pulsating balloon introduced in a minimally invasive way via left subclavian vein to the pulmonary artery and fixed within it with a nitinol stent. The balloon is inflated and deflated due to the gas reservoir regulating the gas flow between it and the balloon by means of the connecting conduit. The drive unit is implanted under the skin in the left subcostal or subclavian area. It does not require an external power supply [43]. Its pictorial drawing is presented in Figure 2(4). 

The balloon inflates during the right ventricle diastole and deflates in the right ventricle systole. It mimics the functions of healthy vessels and as a result improves the compliance of the pulmonary vasculature. The device simultaneously reduces the workload of the right ventricle while increasing the right ventricular output. The hemodynamic effects described above and the safety of using this system must be proven in prospective, randomized trials.

### 4.3. Right Ventricular Pacing

Another device with hopes of improving right ventricular function in PAH patients is right ventricular pacing. The idea of its application stems from observations from echocardiographic tests (RV strain) and magnetic resonance imaging studies [44]. These studies have shown that patients with PAH exhibit intraventricular and interventricular dyssynchrony during systole that it is more mechanical than electrical in nature. Earlier studies on an animal model and modeling with computer algorithms, as well as acquired observations from the use of RV pacing in patients suffering from CTEPH, suggest a beneficial effect of this method of treatment [45,46]. 

In order to answer the question whether this procedure would improve the hemodynamics of the right ventricle and what consequences it would have in terms of energy expenditure and stresses on the right ventricular walls, a study of Right Ventricular Pacing in Pulmonary Arterial Hypertension (ClinicalTrials.gov Identifier: NCT04194632) was started on 1 January 2021 [47]. Upon completion of the study scheduled for 5 December 2021, responses to the endpoints including the impact of RV pacing on change in contractility, change in stroke volume, pressure-volume loop area, and global myocardial oxygen consumption should be known.

### 4.4. Extracorporeal Membrane Oxygenation

Patients with severe PAH experience RHF and left heart insufficiency at the end of this disease. The previously mentioned methods of treatment may improve or at least maintain patients’ conditions, but all of them have only limited feasibility as a means of therapy. ECMO is a valuable option for such patients. There are two types of ECMO: VV-ECMO used for respiratory failure and veno-arterial ECMO (VA-ECMO) implemented in patients with cardiorespiratory or cardiac insufficiency. For those with PAH, VA-ECMO was found to be applicable. Its scheme and exemplary photographs are presented in Figure 3.

#### 4.4.1. Methods of Extracorporeal Membrane Oxygenation Implantation

(a)Surgical: This method is the most common method of ECMO implementation, which is based on the preparation and visualization of the targeted vessels followed by cannulation. Usually, the working part of the venous cannula (drainage site) should be positioned within the right atrium and superior vena cava under ultrasound supervision (preferably transesophageal echocardiography). One must not forget about peripheral perfusion of the leg (PLP) (Figure 3(1-E)), which is obtained as a connection between the arterial cannula and the distal part of the femoral artery. Rarely, when Willis circulation is sufficient, there is a possibility of placing an arterial cannula through the right common carotid artery surgically and placing a venous cannula in the right internal jugular vein (IJV) so the patient can be treated not only awake but also mobilized [48]. This type of implantation can be used for intermediate- and long-term ECMO support.(b)Percutaneous: The Seldinger method used for cannulation of the right IJV and/or inferior vena cava as well as the femoral artery. If intended for short-term support, a PLP shunt may be omitted in exceptional circumstances, and strict supervision of the lower extremity is applied.(c)Mixed: This method includes the cannula to right IJV with the Seldinger method and surgical insertion of the arterial cannula (through the femoral artery into the distal part of the abdominal aorta) and the PLP cannula. This type of implantation can be used in patients requiring intermediate- and long-term ECMO support.

#### 4.4.2. Purpose of Extracorporeal Membrane Oxygenation Implantation

##### Ultimate Bridge to Lung or Heart-Lung Transplantation

ECMO has its application as a life-sustaining device in the face of the exhaustion of all other therapeutic options among certain patients. PAH may eventually cause deterioration of the patient’s condition to the point of thoracic organ transplantation being the only viable treatment. However, the global issue of scarcity of lung donors as well as organ donors in general may require some qualified patients to be bridged to the lung or heart-lung transplantation (HLTx) [49,50]. This was the case reported by Gregoric et al., who described a young patient with IPAH requiring emergency HLTx after unsuccessful RVAD implantation [51]. The patient was bridged with VA-ECMO and became an HLTx recipient after 10 days of support.

Moreover, the initial cause of the deterioration was suspected to be massive pulmonary embolism. The authors of this paper also reported the case of a patient who suffered from pulmonary hypertension in group 4 and was bridged to Ltx for 90 days (one of the longest supports in Europe) for the exact same reason: rapid deterioration of his condition. Our patient was only a right single lung recipient due to preexisting paralysis of the left phrenic nerve. This was the first patient to be successfully bridged via ECMO in Poland [52]. As high mortality among patients with PAH is associated with RHF followed by secondary multiorgan failure, proper treatment and transplant eligibility should be implemented at the right time.

Due to limitations of donor thoracic organs, some patients with PAH may develop complications, such as hypoxemic liver failure and pre-renal renal failure. While waiting for the lung/heart-lung graft [53]. VA-ECMO is the ultimate bridging therapy for such patients. It can potentially reverse these complications to a certain extent by providing better systemic circulation and oxygenation. One must bear in mind that VA-ECMO is also associated with adverse effects, with bleeding being the most common and frequent adverse cerebrovascular event, which usually occurs two weeks after initiation of such therapy [54,55]. It is worth mentioning that such bridging to Ltx is not associated with worse outcomes after Ltx, provided expert centers are performing such treatment [56].

##### Perioperative Support and Left Ventricular Conditioning

Short-term results of patients with IPAH reveal the highest risk of dying and complications during the early period after Ltx. The ISHLT Registry showed that even though patients with IPAH have the second-best long-term survival (12.0 years conditional on 1-year survival), they also have the highest short-term mortality (25% at 1-year) [57]. Hoetzenecker et al. reported that patients with intraoperative use of ECMO as cardiorespiratory support showed improved long-term survival when compared with patients operated on without such support (91% vs. 82% reached 1-year survival, 85% vs. 76% reached 3-year survival, and 80% vs. 74% reached 5-year survival; log-rank *p* = 0.041) [58]. Their results were still valid after propensity score matching for both cohorts. There was also a comparison between ECMO and cardiopulmonary bypass (CPB) as cardiopulmonary support during transplantation. Intuitively, the aim of such a procedure is to avoid CPB due to higher activated clotting time values, which are associated with an increased risk of bleeding. The perioperative blood product transfusion requirement was lower in the ECMO group than in the CPB group, as reported by Machuca et al. [59]. Patients from the ECMO group presented a significantly shorter duration of mechanical ventilation, length of intensive care unit stay, and hospital stay (median 3 vs. 7.5 days, *p* = 0.005; 5 vs. 9.5 days, *p* = 0.026; 19 vs. 27 days, *p* = 0.029, respectively). VA-ECMO was also proven to be superior to standard CPB as well as the modality of choice during Ltx. The authors of this review also presented results consistent with the aforementioned studies as all of the patients transplanted due to end-stage PAH were treated with intraoperative use of VA-ECMO [21]. In most cases, ECMO was introduced as described in the previous paragraph entitled Methods of ECMO implantation, point a).

As gradually increasing pulmonary vascular resistance can lead to RHF in patients with primary pulmonary hypertension, the right ventricle becomes hypertrophic in response to the described phenomenon. Ltx provides pulmonary vessels with healthy properties; hence, RV hypertrophy will work in the lung recipient’s disadvantage. Impaired functions can lead to RV outflow tract obstruction, and the LV will experience a sudden increase in preload after Ltx. However, as the left ventricle was not required to work with such preload for a long period before transplantation, it may be dysfunctional enough for the patient to develop pulmonary edema and primary graft dysfunction. It should be avoided by proper conditioning of the heart, achieved by intra- and postoperative use of VA-ECMO. The general rule is to unload the left ventricle and then gradually increase the native cardiac output of the left ventricle by diminishing ECMO support. Each change in the extent of ECMO support was performed under echocardiographic supervision. This approach has been applied to lung transplant recipients in the lung transplant facilities of Vienna and Hannover [58,60,61,62,63]. However, there were differences between the groups. The Viennese team used prolonged mechanical ventilation and postoperative VA-ECMO therapy in sedated patients [60,63]. Patients with prolonged VA-ECMO support had a 5-year survival rate of 87.4%, according to Moser et al. [60]. However, the lung transplant protocol in Hannover regarding prolonged postoperative VA-ECMO heart conditioning favors early extubation and the patient is awake during this time [61,62]. Tudorache et al. reported that survival rates at 3 and 12 months among 23 patients were 100% and 96%, respectively [62]. Salman et al. published a study in Germany that revealed that patients who underwent double Ltx followed by proper VA-ECMO-based left heart conditioning can reach survival comparable to those who were transplanted due to other end-stage lung diseases [61]. They also reported favorable hemodynamic changes (the proximal RV outflow diameter decreased from a median of 48 mm to a median of 25 mm, *p* < 0.05, and LV end-diastolic dimension increased from 40 mm to 45 mm, *p* < 0.05), which validated this treatment as the aforementioned conditioning tool.

Comparable data were reported in our previous study [21]. The 30-day survival rate was 100% in our patients. These results can contribute to proper LV conditioning. One of the patients underwent a concomitant procedure as he developed severe tricuspid regurgitation due to right heart insufficiency that had to be addressed. For this procedure, we used our technique of incorporating the use of ECMO and CPB during the same complex procedure and achieved good outcomes with respect to 30-day mortality [21].

## 5. Thoracic Organ Transplantation

Ltx and HLTx remain the only viable surgical treatments for patients refractory to either the pharmacological or interventional treatments described above [54]. The organs procured for such procedures are depicted in Figure 4. Currently, careful selection of patients provides satisfactory results. Key decisions involve the choice of the transplantation type. Both types of transplantation restore proper vascular resistance to the pulmonary vessels, making interventional procedures such as AS and Potts shunting (if previously performed) unnecessary if not harmful after transplantation. Paradela et al. described the case of a female patient, who, during one complex procedure had her Potts shunt closed with the aorta being sewn over the endovascular stent-graft followed by immediate double Ltx. Left pulmonary artery anastomosis was performed proximal to the Potts shunt insertion. She was discharged in a good general condition [64]. In the aforementioned case report by Prieto et al., a butterfly stent located in the atrial septum was used as a bridging modality in patients awaiting transplantation [23]. It was removed during the procedure of bilateral Ltx, as it was decided that leaving it there would not be beneficial and dangerous. Our department also experienced a patient with AFR, which remained intact during the operation and early period afterward. Systematically performed echocardiography during the postoperative period revealed no flow through the device. He was discharged in a good general condition and remained under the supervision of the cardiology department; the decision of whether to explant or close it will be made later [20,21] There is an issue if the AFR and AS remain intact after Ltx, as these factors increase the risk of neurological complications such as stroke. The question remains whether persistent left-right shunt after Ltx increases the risk of PAH recurrence. We acknowledge the importance of postoperative echocardiography and a long-term follow-up.

### 5.1. Lung Transplantation

This therapeutic option should be discussed for patients with NYHA functional classes III and IV refractory to the available maximal dosage of pharmacological treatment [65]. Increasingly better survival rates after Ltx have been reported yearly among patients with IPAH. This particular underlying disease has the second-best survival among all patients treated in this manner (cystic fibrosis being the best one). According to the Registry of International Society of Heart and Lung Transplantation, 50% of patients survive for 7 years. As double Ltx provides superior survival in comparison to its single lung counterpart (50% survival over 7.8 years for double lung transplant [DLT] vs. 4.8 years for SLT, *p* < 0.0001), it is generally advised to perform bilateral Ltx [57]. Additionally, there is an increased mortality hazard among SLT recipients with an mPAP exceeding 40 mmHg, as reported by Nasir et al. [66]. Median survival of Ltx recipients due to PAH was 12 years among those who reached at least 1-year survival after transplantation [57]. It is worth noting that solid organ transplantation is performed only in patients with no other therapeutic options. As careful candidate selection and proper time of the transplant center referral and listing provide good outcomes of the procedure, facilities treating patients with PAH should be familiar with guidelines pertaining to thoracic organ transplantation [67]. There are some concerns about immunosuppressive treatment, which has certain adverse effects, such as an increased risk of infection and carcinogenesis [68]. VA-ECMO plays an important role not only as a bridge for Ltx, but also as a cardiopulmonary support of choice during such procedure, and its prolonged postoperative use provides needed heart conditioning [59,60,61,69].

### 5.2. Heart-Lung Transplantation

Such treatment is performed in patients with PAH and significant, irreversible right (or biventricular) heart failure. The first recipient of HLTx was an American woman with primary pulmonary hypertension operated on by a team led by Reitz [70]. This multiorgan transplantation is preferred in some thoracic transplant centers among patients with refractory right ventricle failure [54]. Fifty percent of patients after HLTx reached survival of 5.5 years [57]. The same parameter among patients who underwent such procedures and survived for at least 1 year was approximately 10.7 years. In this aspect, our facility has some experience, as the longest survival after such a procedure among our patients was 19 years [71]. The patient remained under our follow-up care. The results of singular Ltx were worse than those reported after bilateral Ltx. However, survival after HLTx has improved over the years. Brouckaert et al.’s study reported that among recipients with double Ltx and HLTx, overall 5-year graft survival was comparable (70% for DLT recipients vs. 61% for HLT recipients) [72]. Primary graft dysfunction among patients was lower after HLTx than after double Ltx (76.7% vs. 23.8%). Nonetheless, double Ltx remains the preferred procedure for all forms of precapillary pulmonary hypertension, except in patients with severe congenital heart disease. Comparable survival among patients with IPAH treated with either double Ltx or HLTx was also reported by Hill et al. [73]. Between 1991 and 2000, 90% of thoracic organ transplantations occurred among patients with severe pulmonary hypertension because the technical feasibility of performing double Ltx was low. Recent data from 2011–2014 reported that this form of treatment decreased to 20% among the same group of patients. This finding was connected to the fact that the results of Ltx have significantly improved over the years. Then, the need for the HLTx procedure was diminished concomitantly, enabling more heart transplantations as the availability of these organs increased.

## 6. Conclusions

In conclusion, PAH requires an interdisciplinary approach using pharmacological, interventional, and surgical modalities. It was established that in patients with severe PAH, the aim is to transfer the suprasystemic form of this disease into Eisenmenger physiology, as it is associated with improved survival. AS and the Potts shunt can be useful bridging tools for Ltx, which remains the final surgical treatment among patients who are refractory to any other kind of therapy. VA-ECMO remains the ultimate bridge, cardiorespiratory support during operation, and perioperative LV conditioning therapy for patients with severe PAH. Novel techniques such as RVAD and pulmonary denervation may have potential as emerging new ways of ensuring transplant-free survival among patients with the aforementioned disease. However, high-quality randomized controlled trials are needed to confirm this.

## Figures and Tables

**Figure 1 jcm-10-03326-f001:**
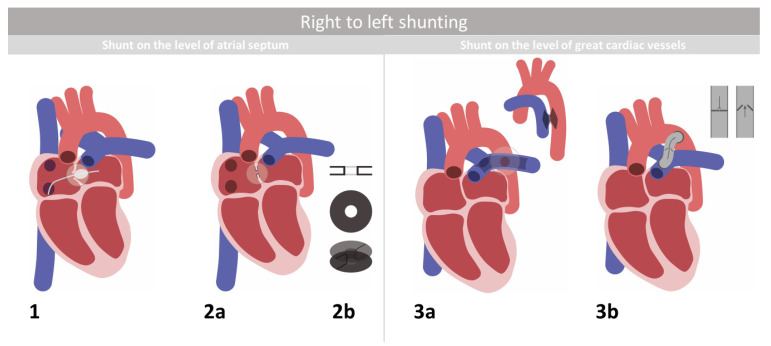
Treatments utilizing right-to-left shunting: (**1**) Atrial septostomy, (**2a**) atrial flow regulator (AFR) at the site, (**2b**) scheme of the AFR, (**3a**) Potts shunting, and (**3b**) unidirectional valve conduit as a modification of Potts shunting.

**Figure 2 jcm-10-03326-f002:**
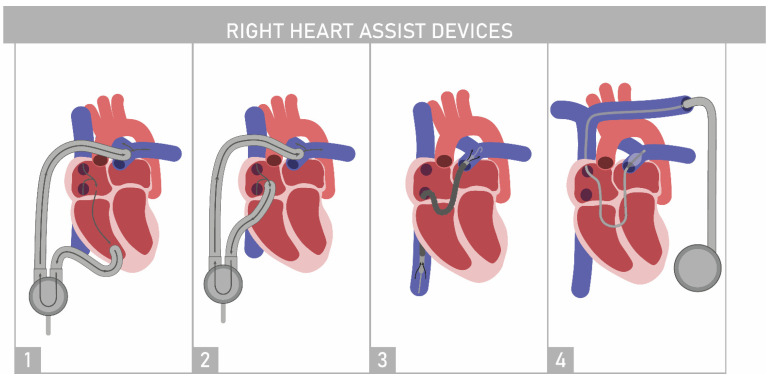
Right heart assist devices: (**1**) right ventricular assist device with right ventricular sourcing, (**2**) right ventricular assist device with right atrial sourcing, (**3**) the Impella RP heart pump (Abiomed, Danvers, MA, USA), and (**4**) the Aria CV PH system (Aria CV, Inc. Saint Paul, MN, USA).

**Figure 3 jcm-10-03326-f003:**
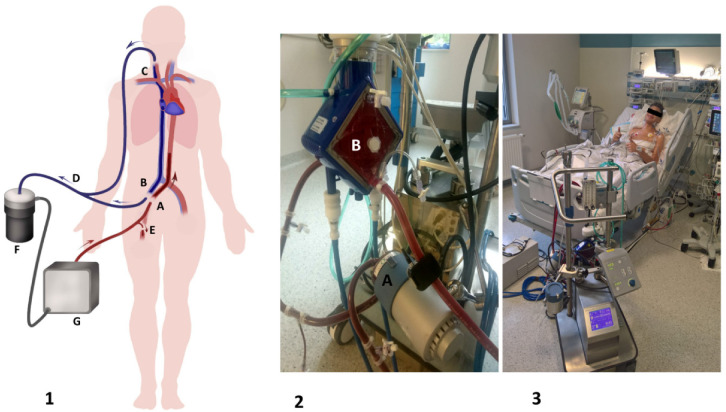
(**1**) Scheme of the veno-arterial extracorporeal membrane oxygenation circuit. (1-A) An inflow cannula is inserted through the femoral artery. (1-B) An outflow cannula is inserted through the femoral vein. (1-C) An outflow cannula is inserted through the right internal jugular vein. (1-D) The common outflow cannula, comprising (1-B) and (1-C), is merged with a Y-shaped connector. (1-E) The peripheral leg perfusion cannula. (1-F) The centrifugal pump. (1-G) The oxygenator. (**2**) Close-up of the pump (2-A) and oxygenator (2-B). (**3**) An awake, extubated patient after double lung transplantation due to idiopathic pulmonary arterial hypertension (authors’ own material).

**Figure 4 jcm-10-03326-f004:**
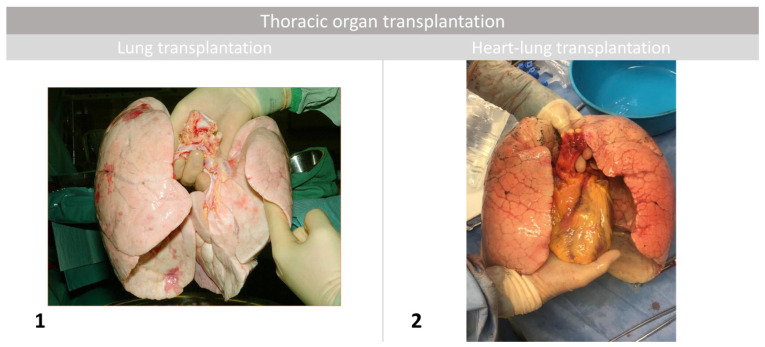
(**1**) Organ (double lung) for transplantation. (**2**) Organs en bloc (heart and lungs) for heart-lung transplantation.

## Data Availability

All of the cited data in this review has come from professional medical literature cited in the Reference section according to order of appearance. Data that describe the experience of this review’s authors have also been properly cited, as they come from our peer-reviewed, published articles and book chapters.

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
