# Peer review of "Interventional and Surgical Treatments for Pulmonary Arterial Hypertension"

_jcm, 2021, doi:10.3390/jcm10153326_

Round 1

Reviewer 1 Report

Thank you for allowing me to review this paper. The paper by Stacel et al is a review on interventional and surgical treatments for pulmonary hypertension.

English form requires a little editing to improve readability. Some typographic errors should be fixed. I have only some minor suggestions:

line 37: …”calcium blockers”: please correct with the current term “calcium channel blockers”. Moreover, please specify the marginal role of this class of drugs in the current management of PAH.

Line 125 …”6-minute walk distance (6 MWT)”: please correct the terms used. The 6 minute walk distance is an item (maybe the main) within the 6 minute walk test, which comprises several items. In the whole paper, authors should clearly define when talking about the distance and when talking about the test.

Author Response

Dear Reviewer, 

Thank you very much for your input. We are grateful that you found time to review our work. Your remarks has been noted and we changed those things that were mentioned.  In order to ensure proper English of the article, we sent our paper for editing to the company providing such services. We will attach the certification if requested.

We attached the changed manuscript.

We hope You're pleased with changes we applied.

Kindest regards, the authors

Reviewer 2 Report

Thank you for allowing me to review Staçel et al.’s article entitled “Interventional and surgical treatment for pulmonary arterial hypertension.” This review summarizes interventional and surgical approaches used in PAH including atrial septostomy, atrial flow regulator, Potts shunt, pulmonary artery denervation, RVAD, VA-ECMO, and transplant. This is an unbiased review that nicely summarizes the currently available interventional/surgical approaches for PAH. I have several suggestions to improve the impact of the review and readability of the article:

  1. There are other interventional procedures currently under investigation for PAH including the Aria CV device (NCT04555161) and RV pacing (NCT04194632). Including these other procedures would allow this review to be more updated and distinguish it from other reviews outlining interventional/surgical treatment options in PAH.
  2. Please include p-values for numerical comparisons included in the text if they are available from the primary studies. P-values were included for some studies described in the text but not most.
  3. Consider changing the presentation of the RVFAC and TAPSE data in lines 164-166 to absolute increases instead of percentage. This can be confusing as RVFAC is reported as a percent.
  4. There are several areas of the text where references are not included, sometimes after listing a study author’s last name. The reference number should still be included after mentioning the author (e.g. lines 481-483). Also, references should be included after each sentence that is stating data from a prior study (e.g. lines 121-125, lines 217-222, etc.).
  5. Please cite a reference describing for the following sentence in lines 182-184: “This is a notable difference in comparison with the AS procedure, which requires tricuspid valve regurgitation, so atrial shunting is possible.”
  6. In the case discussed in lines 295-299, please clarify if this is group 3 PH (due to idiopathic pulmonary fibrosis) and not PAH? Please correct the use of “PAH” to “pulmonary hypertension” if needed.
  7. Please include the clinicaltrials.gov NCT number for the phase II trial investigating PADN (lines 235-237).
  8. For line 240, please describe over how long of a time period did “60% of the patients had their condition deteriorated”.
  9. Please define CPB – suspect it is cardiopulmonary bypass but this was not defined previously.
  10. Recommend discussing global donor scarcity, instead of just in Poland.
  11. Please clarify/reword these lines as the meaning are unclear: Lines 264-265 “As pulmonary hypertension leads to RV failure, mechanical support reduces stroke.”

Line 460 “Conditional to 1-year survival after Ltx was 12 years.”

  1. There is inconsistent formatting of the references in the bibliography.

Author Response

Dear Reviewer, we are glad and grateful that you found time to review our paper. We believe that your remarks made our paper better. To ensure proper english in our article, we used professional services in this regard and attached the proper certificate. All the changes to the original manuscript have been marked with yellow background. We are glad that you generally liked our paper. Here are our answers to your remarks:

  1. There are other interventional procedures currently under investigation for PAH including the Aria CV device (NCT04555161) and RV pacing (NCT04194632). Including these other procedures would allow this review to be more updated and distinguish it from other reviews outlining interventional/surgical treatment options in PAH.

Ad 1. Thank you for this remark. We conducted additional research and added seperate paragraphs for Aria CV device as well as for RV pacing. In our opinion this will trully enrich our article. Additionally, we added pictorial drawing to our figure to improve the understanding of Aria device for the future readers.

2. Please include p-values for numerical comparisons included in the text if they are available from the primary studies. P-values were included for some studies described in the text but not most.

Ad 2. We have added missing p values in all of the remaining places, wherever p value was provided in the original paper

3. Consider changing the presentation of the RVFAC and TAPSE data in lines 164-166 to absolute increases instead of percentage. This can be confusing as RVFAC is reported as a percent. 

Ad 3. This part was changed in accordance to the remark.

4. There are several areas of the text where references are not included, sometimes after listing a study author’s last name. The reference number should still be included after mentioning the author (e.g. lines 481-483). Also, references should be included after each sentence that is stating data from a prior study (e.g. lines 121-125, lines 217-222, etc.). –

Ad 4. That was corrected in the aforementioned places, as well as in other places, which were missing one of aforementioned things. 

5. Please cite a reference describing for the following sentence in lines 182-184: “This is a notable difference in comparison with the AS procedure, which requires tricuspid valve regurgitation, so atrial shunting is possible.” 

Ad 5. Reference was provided, as requested

6. In the case discussed in lines 295-299, please clarify if this is group 3 PH (due to idiopathic pulmonary fibrosis) and not PAH? Please correct the use of “PAH” to “pulmonary hypertension” if needed. 

Ad 6.  Correction was made, as requested

7. Please include the clinicaltrials.gov NCT number for the phase II trial investigating PADN (lines 235-237).

Ad 7. Upon searching, we were not able to locate the NCT number for aforementioned paper. However authors of this paper provided the number for clinical trial registration  URL: http://www.chictr.trc.com.cn. Unique identifier: chiCTR-ONC-12002085. All of this information is available at following page: https://www.ahajournals.org/doi/10.1161/CIRCINTERVENTIONS.115.002837

8. For line 240, please describe over how long of a time period did “60% of the patients had their condition deteriorated 

Ad 8. Correction was made, as requested

9. Please define CPB – suspect it is cardiopulmonary bypass but this was not defined previously. 

Ad 9.  Correction was made, as requested

10. Recommend discussing global donor scarcity, instead of just in Poland.

Ad 10. Correction was made, as requested. Two papers pertaining the issue were cited. 

11. Please clarify/reword these lines as the meaning are unclear: Lines 264-265 “As pulmonary hypertension leads to RV failure, mechanical support reduces stroke ???.”

Line 460 “Conditional to 1-year survival after Ltx was 12 years.”

Ad 11. Both sentences were rewritten, as requested

12. There is inconsistent formatting of the references in the bibliography.

Ad 12. Corrections were made, as requested

We believe that your input has made our work better. We hope that you will be pleased with changes we applied.

Kindest regards, the authors

Round 2

Reviewer 2 Report

The authors were very responsive to my comments. Thank you. Please consider making the following minor revisions when editing the manuscript for publication.

  1. The following sentence in the abstract appears to be missing a word - “As established, obtaining the Eisenmenger physiology among patients with severe [missing a word here – PAH?] by creating artificial defects is associated with improved survival.”
  2. In line 110-112: The patency of AS can be improved by inserting 110 stents or special devices, such as atrial flow regulators (AFRs), presented in Figure 1-2a, 1-2b, and 1-3a. Check the Fig 1-3a reference – should this be Fig 1-3b or not included?
  3. Line 123 – please include additional decimal places or write p<0.01 instead of p=0.00.